# Child and Adolescent Mental Health during the COVID-19 Pandemic: Challenges of Psychiatric Outpatient Clinics

**DOI:** 10.3390/healthcare11050765

**Published:** 2023-03-06

**Authors:** Mariela Mosheva, Yael Barzilai, Nimrod Hertz-Palmor, Ehud Mekori-Domachevsky, Asia Avinir, Galit Erez, Noa Vardi, Gila Schoen, Tal Lahav, Hadar Sadeh, Michal Rapaport, Chen Dror, Alex Gizunterman, Shlomit Tsafrir, Doron Gothelf, Yuval Bloch

**Affiliations:** 1The Child Psychiatry Division, Edmond and Lily Safra Children’s Hospital, Sheba Medical Center, Tel Hashomer, Ramat Gan 5262000, Israel; 2Department of Psychiatry, Sackler Faculty of Medicine, Tel Aviv University, Tel Aviv 69978, Israel; 3The Emotion-Cognition Research Unit Center, Shalvata Mental Health Center, Hod-Hasharon 45100, Israel; 4MRC Cognition and Brain Sciences Unit, University of Cambridge, Cambridge CB2 1TN, UK; 5Geha Mental Health Center, Petah Tikva 49100, Israel; 6Ness Ziona Mental Health Center, Ness Ziona 74100, Israel; 7Soroka Medical Center, Be’er-Sheva 84101, Israel; 8Goldman School of Medicine, Ben-Gurion University of the Negev, Be’er-Sheva 84105, Israel; 9Lev Hasharon Mental Health Center, Natanya 42100, Israel; 10The Jerusalem Mental Health Center, Jerusalem 91060, Israel; 11Clalit Health Services, Jerusalem 91060, Israel

**Keywords:** outpatient clinic, child and adolescents, mental health, COVID-19, child and adolescence psychiatry

## Abstract

Background: Worldwide national surveys show a rising mental health burden among children and adolescents (C&A) during COVID-19. The objective of the current study is to verify the expected rise in visits to psychiatric outpatient clinics of C&A, especially of new patients. Methods: a cross-sectional study focusing on visits as recorded in electronic medical records of eight heterogeneous C&A psychiatric outpatient clinics. The assessment was based on visits held from March to December of 2019 (before the pandemic) in comparison to visits held in 2020 (during the pandemic). Results: The number of visits was similar for both periods. However, in 2020, 17% of the visits used telepsychiatry (N = 9885). Excluding telepsychiatry reveals a monthly decrease in traditional in-person activities between 2020 and 2019 (691.6 ± 370.8 in 2020 vs. 809.1 ± 422.8 in 2019, mean difference = −117.5, t (69) = −4.07, *p* = 0.0002, Cohen’s d = −0.30). Acceptation of new patients declined during 2020, compared to 2019 (50.0 ± 38.2 in 2020 vs. 62.8 ± 42.9 in 2019; Z = −3.12, *p* = 0.002, r = 0.44). Telepsychiatry was not used for new patients. Conclusions: The activity of C&A psychiatric outpatient clinics did not rise but was guarded due to the use of telepsychiatry. The decline in visits of new patients was explained by the lack of use of telepsychiatry for these patients. This calls for expanding the use of telepsychiatry, especially for new patients.

## 1. Introduction

Parallel to worldwide morbidity and mortality, the COVID-19 pandemic has been causing significant emotional distress, pointing attention toward assessing and treating mental health issues. Accumulating reports address the severe and multifaceted consequences on the mental health of children and adolescents, identifying them as a particularly vulnerable group [1]. For example, a study from China conducted at the beginning of the pandemic outbreak found that one third of 3- to 18-year-old children and adolescents were clingy, inattentive, irritable, and worried during the pandemic [2]. Other worldwide studies have reported severe levels of psychological distress such as worries, helplessness, fear [3,4,5,6], and high anxiety and depressive symptoms among children and adolescents during the pandemic [5,7,8,9]. Moreover, recent nationwide studies in the U.S. reported worsening children and adolescents’ psychological well-being and behavioral health compared to before the pandemic [10,11]. Additionally, in Israel, a retrospective cohort study utilized a large, computerized database and found a rise in the incidence of depression, anxiety, and eating disorders, and a rise in the use of antidepressants and antipsychotics during the pandemic years [12]. It is of note that most of these studies were based on community surveys and questionnaires that reflected a rise in emotional distress, but not necessarily an increase in psychopathology or treatment-seeking. While hotlines, community support, and counselors relate to distress and not to psychopathology, treatment in psychiatric outpatient clinics requires a professional diagnosis.

The extent to which pediatric mental health services are affected by COVID-19 and its long-term impact is still under investigation. A study that investigated hospitalization numbers in youth psychiatric wards in Israel found that the number of hospitalized patients decreased in 2020, compared to 2019 [13]. Studies assessing ER (Emergency Room) visits in the U.S. showed a marked decrease in pediatric ER visits across a broad range of conditions; however, the proportional decline in mental health visits was less pronounced [14]. Moreover, patients with mental health conditions presenting for ER visits since the onset of the pandemic were more likely to require admission and have had more prolonged admissions [15]. A recent study focusing on the first year of the pandemic exemplified a decline in psychiatric ER visits of C&A, especially those suffering from stress-related anxiety and depressive disorders [16,17]. In addition, C&A that did not have a previous encounter with mental health outpatient clinics were less likely to visit the ER in 2020. The burdens of lockdowns, quarantines, and social distancing, which typified the first year of the pandemic, are all suggested as major stressors for the pediatric population [18]. The outpatient clinics are probably less intimidating than the psychiatric ER or the general ER, especially at the time of the pandemic. One would expect a rise in the activity of outpatient clinics that are central to evaluating and treating stress-related anxiety and depression disorders in the community.

To overcome the challenges of physically visiting the outpatient clinics during the pandemic, the use of telemedicine and telepsychiatry (the application of telemedicine within the specialty of psychiatry) was suggested [19]. Telepsychiatry is the delivery of psychiatric or mental health services via telecommunications technology, usually video. Studies aimed to investigate if telepsychiatry is a comparable tool for assessment and treatment show patients and providers are usually satisfied with telepsychiatry. Moreover, it was found that telepsychiatry is a comparable tool to face-to-face service in terms of reliability of assessment and treatment outcome [20,21]. However, before the pandemic, outside the research setting, the use of telepsychiatry was slow and limited by clinicians’ concerns about regulation and quality of care [20,21]. Resulting from the worldwide COVID-19 crisis, the use and implementation of telepsychiatry has increased [19,22,23,24]. A few recent papers studied outpatient clinics in different countries and displayed a decrease in referrals to psychiatry outpatient clinics compared to the year before the onset of the pandemic [25,26]. Nevertheless, to the best of our knowledge, no studies have examined the actual effect of the pandemic on psychiatric outpatient mental health referrals among children and adolescents in Israel, compared to the year before the pandemic onset. The current study aims to examine the real-life activity of psychiatric outpatient clinics during the COVID-19 pandemic (March to December 2020) compared to the comparator period in the previous year (2019). Based on literature pointing to an increase in mental health burden, we hypothesized that there would be a rise in mental health needs and therapy, specifically for new patients. Beyond the general trend and numbers, it is important to evaluate the effects on different therapeutic encounters. During the pandemic, some therapeutic encounters, such as group therapy, are expected to be more challenging due to the need for social distancing. It is important to study the applicability of telepsychiatry in mending the gap between the rising needs and difficulties in accessibility to outpatient visits. Studying multiple outpatient clinics that differ in the population they serve and evaluating the condition, not just during quarantine, can aid in controlling for some of the confounders. Thus, we aimed to collect the data from eight outpatient clinics with different characteristics.

## 2. Materials and Methods

### 2.1. Sample

The Israeli population is heterogeneous in ethnocultural origin, religion, and type of residency. Thus, it probably allows for better generalization compared to different populations. During the pandemic, there were more cases of infection among the Muslim minority and the Ultraorthodox Jewish community in comparison to the general Israeli population (possibly partially due to less awareness of the dangers of the pandemic and more suspiciousness about regulations set by the government) [17,27]. In the Israeli health system, therapy in outpatient clinics is free of charge and provided by the four large health providers. While some patients prefer private therapy, we estimate that most C&A are treated in the public system. Approximately two percent of the general child and adolescent population is served by the public mental health system [28].

We conducted a cross-sectional study collecting data from electronic medical records of eight public outpatient clinics in central Israel. Though we do not know the size of the population that was treated in other facilities (private or public) outside these eight outpatient clinics, we present some data on the centers and their catchment area, and we believe a considerable part of the mental health services for C&A are given by these centers. Two of the clinics are located at general tertiary hospitals. Sheba Medical Center is located in the highly populated center of the country. It was a leading center in treating COVID-19 patients, and due to its role as a leading center, can accept patients from all over the country. Soroka Medical Center is located in the south and serves a population of approximately one million inhabitants. Most of this population is considered part of Israel’s “periphery” and includes both rural and non-rural habitats and a large population of Bedouins (a Muslim population with a culture and tradition of its own).

Three clinics are affiliated with sizeable mental health centers (Geha Mental Health Center, Shalvata Mental Health Center, and Ness Ziona centers). The Geha Mental Health Center serves all age groups. It covers a population of approximately 800,000 inhabitants and covers a comparatively large population of Ultraorthodox Jews. The Shalvata Mental Health Center serves all age groups. It covers a population of approximately 500,000 inhabitants in the center of Israel. The catchment area includes a heterogeneous population of both Jews and Muslims. The Ness Ziona Center is the largest psychiatric center in Israel and covers an estimated population of more than a million inhabitants. Three community mental health clinics are affiliated but separated from sizeable mental health centers (community clinics affiliated with Lev Hasharon, Shalvata, and Geha). Lev Hasharon Mental Health Center serves a diverse population due to its location near both orthodox and Muslim communities, and serves approximately 500,000 inhabitants.

The data analyzed covered two periods. 1 March to 31 December 2019, and 1 March to 31 December 2020. During that time in 2020, the number of COVID-19 cases in Israel escalated to 434,227 cases (4.71% percent of the population), of them 133,391 (30.71% of all COVID-19 cases) were children and adolescents. This was compared to a comparator period (1 March to 31 December 2019). Since the first case of COVID-19 in Israel was diagnosed on 27 February 2020, January and February of both years (i.e., 2019 and 2020) were removed from the analyses.

### 2.2. Statistical Analysis

Paired *t*-tests were used to compare the number of monthly visits in 2020 and 2019. Each month of 2020 (March–December) was matched with its comparable pair in 2019. For example, monthly visits in March 2020 were compared to the number of visits in March 2019, April 2020 was compared to April 2019, etc. The normality of data distribution was tested using the Kolmogorov–Smirnov (K–S) test for goodness of fit. First, we compared the number of in-person visits (excluding telepsychiatry) and later conducted an additional *t*-test to compare the number of overall visits (in-person + telepsychiatry). We controlled the False Discovery Rate (FDR) with Hochberg and Benjamini’s correction for multiple comparisons to account for type-I errors, and we presented the adjusted *p*-values [29]. The number of no-shows was tested similarly to assess differences between 2020 and 2019. We calculated effect sizes for all of the above-mentioned analyses with Cohen’s d [30,31] and followed the accepted rule of thumb for small (d = 0.2), medium (d = 0.5), and large (d = 0.8) effect sizes [32]. Additionally, we conducted a sensitivity analysis by stratifying the general vs. psychiatric hospital data.

We compared the number of group therapy meetings and new patients’ assessments (intake meetings, the first examination a patient undergoes when admitted to the clinic and held only in person) between 2020 and 2019 using the non-parametric Wilcoxon test for paired samples. We conducted non-parametric testing, instead of the parametric paired *t*-test, due to the data’s positively skewed distribution. We compared group therapies with and without telepsychiatry and performed FDR correction as described. Effect sizes were calculated with rank-biserial *r* correlation.

Analysis was conducted using the *stats* and *rstatix* packages in R version 4.0.3 [33,34].

## 3. Results

### 3.1. Sample Characteristics

From March to December 2020, there were 56,633 visits (13,696 females, 24.2%) to outpatient child and adolescent psychiatry clinics, slightly less than the comparator period in 2019 (n = 58,294). The proportion of children younger than 12 years old decreased from 54.0% in 2019 to 46.0% in 2020, and the proportion of those older than 13 years old increased from 46.0% to 54.0% during the pandemic (Z = −27.1, *p* < 0.0001, Cohen’s h = 0.16). The distributions of age and gender are described in Table 1. Due to technical difficulties, we were able to address diagnostic differences only for patients treated at the Shalvata Mental Health Center.

### 3.2. Type of Visits

The central finding of the current study is that during the first year of the pandemic, a time of extreme need, in-person therapy declined in comparison to the previous year. Paired *t*-test revealed a significant monthly decrease in the traditional in-person activities between 2020 and 2019 (691.6 ± 370.8 in 2020 vs. 809.1 ± 422.8 in 2019, mean difference = −117.5 (−14.5% decrease), t(69) = −4.07, *p* = 0.0002, Cohen’s d = −0.30). However, when telepsychiatry visits, accounting for 17% of the visits in 2020 (N = 9885) were included in the analysis, there was no significant difference in the number of monthly visits between 2020 and 2019 (832.8 ± 385.3 in 2020 vs. 809.1 ± 422.8 in 2019, mean difference = 23.7 (−0.03% decrease), t(69) = 1.19, *p* = 0.24, d = 0.06) (Figure 1). Monthly visits are depicted in Figure 2. There was no difference in the monthly no-show rate between 2020 and 2019 (18.9% vs. 19.1%, t(40) = −0.11, *p* = 0.91, d = −0.02). These trends were consistent during months when lockdowns were imposed (March–April, October–November) and non-lockdown periods. During lockdowns, there was a significant decrease in in-person visits (t(27) = −2.09, *p* = 0.046, d = −0.28) but no differences were present when telepsychiatry visits were taken into account (t(27) = 1.21, *p* = 0.24, d = 0.12) (Figure 3). Similarly, there was a significant decrease in in-person visits during non-lockdown months (t(41) = −3.67, *p* = 0.001, d = 0.31), but this decrease was eliminated when telepsychiatry was included (t(41) = 0.37, *p* = 0.72, d = 0.02). Sensitivity analysis results for the general vs. psychiatric data showed similar trends at the two stratified subsets (Figure 4).

A couple of outpatient therapies were specifically more challenging during the pandemic, mainly group therapy and assessment of new patients. Paired Wilcoxon test revealed a significant monthly decrease in group therapies held in person between 2020 (Median: 282.5, IQR: 243.2) and 2019 (Median: 536, IQR: 360.25). (48.5 ± 70.3 in 2020 vs. 80.8 ± 97.5 in 2019, mean difference = −32.2 (−40.0% decrease), Z = −3.76, *p* = 0.0003, r = 0.49). However, also in this case, online group therapy with telepsychiatry attenuated this decline; thus, the decrease was only descriptive with no significant differences (Median: 72, IQR: 46 in 2020 VS. Median: 0, IQR: 0 in 2019) (60.7 ± 68.1 in 2020 vs. 80.8 ± 97.5 in 2019, mean difference = −20.0 (−24.9% decrease), Z = −1.65, *p* = 0.10, r = 0.20).

We aimed to examine the differences in the encounters coded as “intake” frequencies. The paired Wilcoxon test showed a decrease in the evaluation of new patients during 2020 (Median: 258.5, IQR: 50.75), compared to 2019 (Median: 314, IQR: 53) (50.0 ± 38.2 in 2020 vs. 62.8 ± 42.9 in 2019, mean difference = −12.8 (−20.4% decrease). Z = −3.12, *p* = 0.002, r = 0.44). In line with this, the mean duration of being on the waiting list for an appointment was shorter in 2020 (Median: 3.7, IQR: 1.5) as compared to 2019 (Median: 5.7, IQR: 1.2) (24.6 ± 12.6 days in 2020 vs. 38.7 ± 29.9 days in 2019, mean difference = −14.1 (−36.4% decrease), Z = −2.26, *p* = 0.024, r = 0.50). It is worth noting that data on the waiting list was obtained from only two medical centers (Sheba Medical Center and Shalvata).

## 4. Discussion

The present study aimed to examine the change in therapeutic services in child and adolescent psychiatric outpatient clinics during the first year of the COVID-19 pandemic in Israel. At that time, surveys and publications from non-formal services suggested an increase in the mental health burden of children and adolescents [35]. Several challenges may have contributed to this burden, such as social distancing, fear of infection, reorganization of family routine, and possible loss of family members or friends due to the pandemic [36,37]. Because of those challenges, many researchers and world health agencies (WHO, UNICEF, AACAP, and others) mentioned the need for psychological interventions and supportive care, including early identification of children’s mental health deterioration by pediatric healthcare workers, differentiating symptoms indicating a need for a referral to a psychiatrist, and establishing guidelines to cope with pandemic-related mental health problems [38,39]. Despite those survey results and the many challenges people had to cope with at that time, we found a similar number of psychiatric outpatient visits in 2020 compared to the comparator period in 2019.

There are two reasonable explanations. First, it is possible that the stress caused was evaluated as a natural consequence and that the youngsters did not need professional help from psychiatric services.

The second reasonable explanation would relate to barriers to the possibility of reaching therapy. This can be explained by the fear of visits to medical centers due to fear of being infected, leading to a decline in child and adolescent visits to mental health clinics [40]. However, the current multicenter study included only two centers in a general hospital. The others are psychiatric clinics in psychiatric facilities or in the community, where this fear is probably less likely. In the present study, the referral trends were unchanged in both settings. The more extended period (10 months) studied in the present study included prolonged periods with fewer restrictions related to transportation and a general decline in fear of being infected. Thus, the suggested explanation relating to difficulties in accessibility is probably not the sole explanation.

It has been suggested that the closure of schools, usually considered a burden on mental health, might lower the stress at least temporarily of some vulnerable youngsters, such as those who encounter school bullying [41] or those who suffer from difficulties in academic performance and social problems. Limited direct contact with teachers and counselors, often the first to recognize mental health difficulties, is possibly an additional cause for limited referrals.

The unpredictable nature of COVID-19 and significant uncertainties surrounding short- and long-term treatment strategies resulted in the need for healthcare services to reform and explore alternative modes of service delivery such as telehealth. A recent large-scale study assessed the characteristics and barriers in the transition.

The most striking difference between the two periods is that in 2020, 17% of the visits used telepsychiatry. Remote community treatment and support have long been suggested but have not previously been implemented widely because of challenges to healthcare staff and service users. These include technological barriers, legal, regulatory, and ethical issues [42]. Clinical barriers include specific considerations in assessing emergencies such as suicidality in youth [43], difficulties in communication, and difficulties evaluating young children with disruptive behaviors and developmental disorders [44]. However, in the present study, it is the telepsychiatry that “kept” the service accessible. Those therapies that telepsychiatry did not cover, precisely the encounter with new patients, had a bothering decline. Due to prudence and insufficient awareness of the possibilities of telepsychiatry, we argue that our clinics could not answer to the inclining need of youngsters for therapy. This explains the rise in the use of hotlines and not professional psychiatrists and psychologists. It is important to note that the use of telepsychiatry is also burdened by public acquaintance, comfort, and accessibility to relevant technology. For example, some of the more conservative societies in Israel (ultraorthodox Jews), do not use smartphones and the Internet to the same degree as the secular population.

Evidence supporting the feasibility, acceptability, and effectiveness of telemedicine with children and adolescents in psychiatry is emerging incrementally. Telepsychiatry services in the child and adolescent population have been functioning with promising results [4,43,44,45]. The COVID-19 pandemic has further highlighted the need for provisioning and setting up child and adolescent telepsychiatry services [46], including training medical staff, pharmacotherapies, and psychotherapies. Moreover, a recent large-scale study of eight children and adolescent mental health clinics has shown a rapid pivot from in-person services to home-based telehealth [47].

Online group therapy is a relatively new modality for leading groups. It bears concerns such as psychotherapists’ worries about being less able to communicate their empathy, build therapeutic alliances [48], or worry about the impact of technical barriers and confidentiality issues [49]. As shown in our study and previous studies, with the pandemic outbreak, it became even more crucial to move groups online [50].

## 5. Limitations

Our study has several limitations. First, the retrospective chart review design. It is possible that some of the cases were handled in psychiatric emergencies, outpatient clinics, or private practice. Though we used eight different centers serving a heterogeneous reasonably sized population, some of the information may be missing. The data itself relates to the therapeutic encounters. Other information, and most specifically diagnoses and outcome measures, were not available. In addition, we covered ten months of the pandemic. Thus, some of the effects might emerge as the pandemic continues. Furthermore, we did not include periods of lockdown that may affect the referrals to the clinics.

## 6. Conclusions

There was no rise in visits to psychiatric outpatient clinics of C&A, during the first year of the COVID-19 pandemic. The therapeutic activity was salvaged by the fast incorporation of telepsychiatry. The lack of routine use of telepsychiatry for evaluating and treating new patients is probably the main reason for the decline in this type of therapeutic activity. Thus, our findings highlight the invaluable need for telepsychiatry in treating children and adolescents during crisis periods and social isolation. The results highlight the crucial need for the use of telepsychiatry in evaluating new patients. Future studies should focus on outcome measures beyond the reliability of therapy itself.

## Figures and Tables

**Figure 1 healthcare-11-00765-f001:**
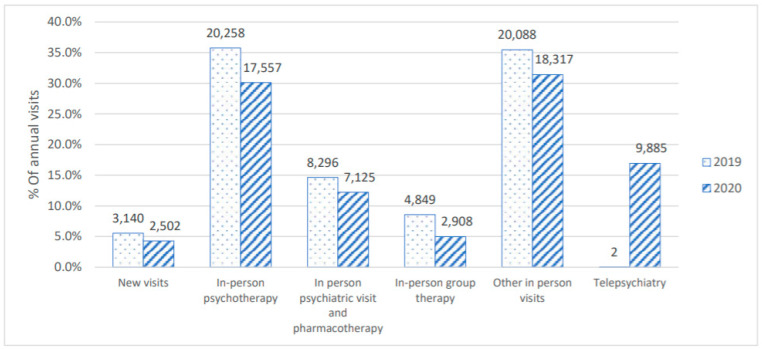
Annual trends in visit type and use of telepsychiatry between 2019 and 2020. Y-axis represents the percent of specific visit types out of all annual visits. Data labels represent the cumulative sum of visit types during each year. No error bars are presented due to the use of cumulative sums instead of means.

**Figure 2 healthcare-11-00765-f002:**
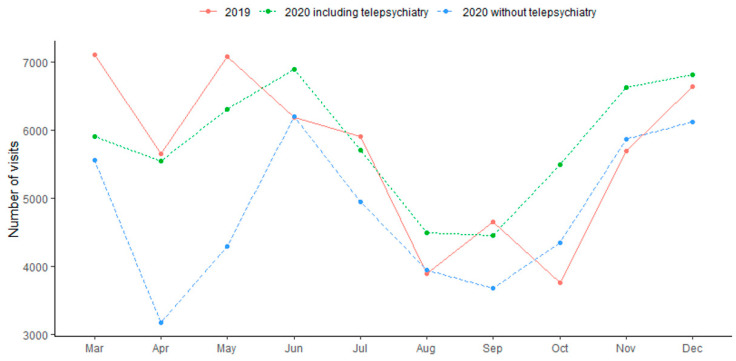
Monthly visits in 2019 as compared to 2020 (stratified with and without telepsychiatry).

**Figure 3 healthcare-11-00765-f003:**
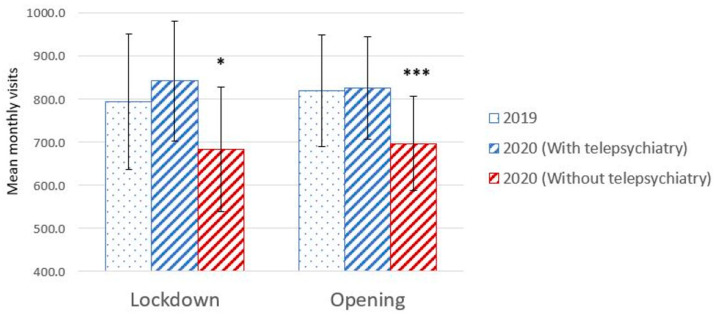
Monthly visits during the lockdown and opening periods in 2019 as compared to 2020. Bars relate to standard deviation. * *p* < 0.05, *** *p* < 0.001.

**Figure 4 healthcare-11-00765-f004:**
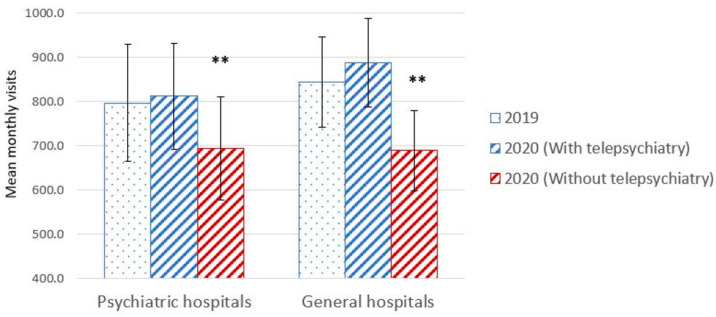
Monthly visits to psychiatric hospitals and general hospitals in 2019 as compared to 2020. Bars relate to standard deviation. ** *p* < 0.01.

**Table 1 healthcare-11-00765-t001:** Demographic variables of patients.

		March–December 2019 Total = 56,663 N%	March–December 2020 Total = 58,294 N%
Gender Male (%)		38,095 (67.27%)	38,442 (65.95%)
Female (%)		18,538 (32.73%)	19,852 (34.05%)
Age groups			
<12	20,310	10,968 (54.0%)	9342 (46.0%)
>13	13,931	6421 (46.0%)	7510 (54.0%)

## Data Availability

The raw data supporting the conclusions of this article will be made available by the authors without undue reservation.

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
