# Peer review of "Child and Adolescent Mental Health during the COVID-19 Pandemic: Challenges of Psychiatric Outpatient Clinics"

_healthcare, 2023, doi:10.3390/healthcare11050765_

Round 1

Reviewer 1 Report

The title is interesting. The idea to compare data before and after pandemic situation is impressive. Authors have analyzed the data very well. however, some feedbacks are provided for the betterment of the manuscript.

Overall. Authors are encouraged to be more focused on the objective. Visualize the study variables so that data collection procedure can be made more clearer. Also, current governmental plan should be described in the text so that the challenges can be explained accordingly in the discussion section.

Abstract:

Objective: Objective of the study should be mentioned

Method: It was secondary data analysis to compare before and after COVID-19 pandemic. Authors are suggested to write clearly about it.

Results:

As authors have not written objective, it is hard to follow what was the findings

Line 30-33:Conducting group therapy and assessing new patients from a distance were criticized. Unlike group therapy, conducted by telepsychiatry, accepting new patients declined during 2020, compared to 2019 (50.0 ± 38.2 in 2020 vs. 62.8 ± 42.9 in 32 2019; Z=-3.12, p= 0.002, r=0.44)”: What was the author’s message and how this result reflect the topic is not clear.

Conclusion: It does not reflect the objective. It is suggested to rewrite in line with the objective.

Introduction:

Even though, there has been mentioned about the need of the study, there still need more justification why this study highlighted specifically child and adolescents towards the challenges of psychiatric outpatient clinics.

Also, introduction of telepsychiatry was one of the treatment plans in most of the countries when pandemic was in peak. Authors are suggested to write this scenario in the introduction section as well.

Line 66-68: The current study aimed to examine the real-life activity of psychiatric outpatient clinics during the COVID-19 pandemic: Does it reflect the challenges of psychiatric clinic? It is suggested to write about the authors understanding about challenges.

Materials and method:

Line 85: The term “sampling period” may not be appropriate as it is secondary data analysis

What were the variables of study? Which variables reflected mental health? How it was recorded?  It should be provided in detail. Thus, data collection procedure should be written to show its reproducibility.

The analysis used are perfect

Results

Type of visits

Line 121-122: “the most striking difference between the two time periods is that in 2020, 17% of the visits used telepsychiatry”. It might be simply because of the current situation due to pandemic. Thus, it is not an exciting result.

Authors only focused on the visit to psychiatric clinic. It would be better if authors could include the data regarding the outcome after the use of telepsychiatry because based on the outcome, it could be suggested for the implementation of telepsychiatry in the day to come in such pandemic situation.

Discussion:

Authors are encouraged to discuss more about the challenges towards mental health of children and adolescents during the pandemic situation with more relevant citations.

Conclusion: should reflect objective of the study

Reviewer 2 Report

Thank you for the opportunity to review this study that aimed to examine the real-life activity of Israeli psychiatric outpatient clinics during the COVID-19 pandemic (March to December 2020) compared to the comparator period in the previous year (2019). This is a large cross-sectional study collecting data from electronic medical records and informing on the role of telepsychiatry in treating children and adolescents during social isolation. Comments to improve the manuscript follow:

Major point

The Authors declared that they aimed to collect the data from eight outpatient clinics with different characteristics (line 74). No context description is provided except for a few lines in the paragraph of the sample. I suggest adding a specific paragraph for the context and providing readers with information showing such differences (e.g., number of population served, extension of the covered area, public/private, ….).

Minor points

Lines 57-60. What does ER mean? Do the authors mean Emergency Department (ED)? Abbreviations should be reported in full the first time they appear.

Line 79. Two. Capital letter not required.

Lines 85-87. “During that time, the number of COVID-19 cases in Israel escalated to 434,227 cases, of them 133,391 children and adolescents”. Adding a percentage may facilitate reading.

In the statistical analysis, the Authors should add Cohen’s d values they referred to with adequate references.

How was the normality of data tested?

Which R version was employed?

Lines 110-112. Remove, not needed.

Table 1. Percentages for gender are missed.

Table 1. I guess there is a typo for age groups: < 12 and >13 (not >12 and <13)

Do the Authors have some more information to describe the two populations? Is the main diagnosis available?

Line 137. “We conducted a sensitivity analysis by stratifying the general vs. psychiatric hospitals data”. This is a method and should be moved to the appropriate section.

Figure 3 and Figure 4. An explanation of the meaning of bars and asterisks is required under the figure.

Lines 142-145 and lines 151-154. Standard differences are substantial. Maybe, the median and interquartile range would have been more adequate measures. I’d suggest presenting both the mean (SD) and median (IQR).

Lines 147-150 provide definitions that should be moved to the methods section.  

Line 157. Remove 3.2. Figures and tables. Place this material next to the text they refer to.

Line 275. Typo. Capital letter required for “our”.

Typo in reference 14.

Round 2

Reviewer 1 Report

Thank you for your effort to correct the manuscript. Now it looks good for the publication. Authors are encouraged to correct typos error through out the text. I have noticed some minor issues to correct as following:

(1) Abstract: “Accepting new patients declined  during 2020, compared to 2019 (50.0 ± 38.2 in 2020 vs. 62.8 ± 42.9 in 2019; Z=-3.12, p= 0.002, r=0.44)”. This sentence is unclear.   Do you mean the "accepting" from the perspective of health care organization? Please rewrite it.

(2) Conclusion: “Thus, our findings highlight the invaluable need for telepsychiatry in treating children and adolescents during crisis periods and social isolation. The results highlight the crucial need for the use of telepsychiatry in evaluating new patients” Suggestion to merge these two sentences as “Thus our findings highlight the invaluable need for telepsychiatry in treating and evaluating new children and adolescents during crisis periods and social isolation”.

Thank you

Author Response

We thank the reviewer, typos were corrected as suggested. 

Reviewer 2 Report

The Authors have adequately addressed all the previous comments. I have no further suggestions.

Author Response

Thank you